# A Bispecific Antibody Blocking Both TSLP and IL-4Rα for the Treatment of Allergic Inflammatory Diseases

**DOI:** 10.3390/cells14221747

**Published:** 2025-11-07

**Authors:** Mingcan Yu, Peng Chen, Ying Jin, Sheng Huang, Hao Jiang, Fulai Zhou, Mark L. Chiu, Di Zhang

**Affiliations:** 1Tavotek Biotherapeutics Inc., 727 Norristown Road, Building 3, Suite 101, Ambler, PA 19002, USA; mingcan.yu@tavotek.com (M.Y.); mark.chiu@tavotek.com (M.L.C.); 2Tavotek Biotherapeutics Inc., 999 Yinshanhu Road, Wuzhong District, Suzhou 215000, China; peng.chen@tavotek.com (P.C.); karen.jin@tavotek.com (Y.J.); sheng.huang@tavotek.com (S.H.); hao.jiang@tavotek.com (H.J.); fulai.zhou@tavotek.com (F.Z.)

**Keywords:** TSLP, IL-4, IL-13, antibody, allergy, antibody engineering, chronic inflammation

## Abstract

**Highlights:**

**What are the main findings?**

**What are the implications of the main findings?**

**Abstract:**

Thymic stromal lymphopoietin (TSLP) works synergistically with Th2 cytokines to regulate infection, inflammation, and metabolic homeostasis. However, their aberrant activities lead to the onset and sustaining of many types of allergic inflammatory diseases. While biologics drug molecules blocking either TSLP or IL-4/IL-13 show clinical efficacies, the broader effect of simultaneously targeting these cytokines remains to be explored. We generated a bispecific antibody (BsAb) targeting both TSLP and IL-4Rα, which effectively blocked the signaling cascades driven by TSLP, IL-4, and IL-13. The BsAb also neutralized TSLP-driven CD4^+^ T cell proliferation as well as IL-4 and IL-13-driven TF-1 cell proliferation. The BsAb reduced CCL17 release from CD14^+^ monocytes activated by LPS, TSLP, and IL-4 and reduced allergen-induced CCL26 and IL-5 release from co-cultures of PBMC, MRC-5, and A549 cells. In a TSLP/OVA-induced asthma model with transgenic human TSLP, TSLP receptor, IL-4, and IL-4Rα mice, the BsAb reduced every single allergic/inflammatory hallmark, while the single target blockade antibody failed to have such comprehensive effects. Our data suggested that simultaneous blocking of TSLP, IL-4, and IL-13 could offer broader control of allergic inflammation, which could translate to a more effective treatment of related disorders.

## 1. Introduction

The pro-inflammatory thymic stromal lymphopoietin (TSLP), an epithelial cell-derived cytokine or alarmin released from airway, skin, and gut epithelia, is critical for mediating allergic Th2 inflammation at barrier surfaces in response to allergens, pathogens, and environmental stressors [1,2]. Upon allergen challenges, the stimulated TSLP primarily promotes myeloid-derived dendritic cell (DC) maturation and proliferation, which in turn primes naïve T helper cells to adopt a Th2 phenotype by producing high levels of Th2 cytokines [3]. These pleiotropic cytokines recruit eosinophilic and basophilic granulocytes and mast cells into the inflammatory site to secrete more inflammatory cytokines and chemokines to establish allergic inflammation [4,5,6,7,8].

Upon binding to a heterodimeric receptor complex composed of the thymic stromal lymphopoietin receptor (TSLPR) and the IL-7R alpha chain (IL-7Rα) [9], TSLP can activate multiple signal transduction pathways that include the activation of Janus kinases (JAKs) which in turn regulate the activity of multiple Signal Transducer and Activator of Transcription (STAT) factors, especially STAT5 [10,11,12]. The TSLP–JAK–STAT5 signaling drives expression of genes that promote DC priming (OX40L, chemokines), Th2 cytokine production (IL-4/IL-5/IL-13), B cell IgE switching, and epithelial remodeling, leading to amplification of type 2 inflammation [13,14]. Besides the JAK/STAT pathway, Src-related tyrosine kinase is involved in TSLP-induced cell proliferation [15].

Among Th2 cytokines, both interleukin-4 (IL-4) and interleukin-13 (IL-13) are critical in orchestrating allergic inflammation [16,17]. They signal through two types of receptor complexes sharing the IL-4 receptor alpha chain (IL-4Rα). The type I receptor complex, composed of IL-4Rα and the common gamma chain (γc), is expressed on hematopoietic cells and exclusively mediates IL-4 signaling [18]. In contrast, the type II receptor complex, composed of IL-4Rα and IL-13 receptor α1 (IL-13Rα1), is broadly expressed on non-hematopoietic cells such as epithelial cells, fibroblasts, and smooth muscle cells, and is activated by both IL-4 and IL-13 [19,20]. A third receptor, IL-13Rα2, binds IL-13 with high affinity but generally lacks classical signaling capacity, functioning primarily as a decoy receptor that limits IL-13–driven inflammation [21].

The engagement of IL-4Rα-containing receptor complexes activates different types of JAKs which leads to phosphorylation and recruitment of STAT6, the central mediator of IL-4/IL-13 signaling [18]. STAT6 drives transcriptional programs required for Th2 differentiation, IgE class switching, and the expression of epithelial genes that recruit eosinophils [22,23]. In addition, IL-4Rα can couple to insulin receptor substrates (IRS-1/2), activating PI3K–AKT pathways that promote survival, metabolism, and alternative (M2) macrophage polarization [24]. MAPK pathways can also be engaged, particularly downstream of IL-13, contributing to airway remodeling and fibrosis [25].

IL-4 is secreted predominantly by activated CD4^+^ Th2 cells, but also by group 2 innate lymphoid cells (ILC2s), basophils, and mast cells [26]. By acting primarily via the type I receptor, IL-4 promotes the differentiation of naïve CD4^+^ T cells into Th2 effector cells and promotes B cell immunoglobulin class switching to IgE [27]. IL-13 is produced by Th2 cells, ILC2s, mast cells, and eosinophils. Signaling via the type II receptor, IL-13 has overlapping functions with IL-4 but is particularly potent in regulating epithelial and structural cell responses, including mucus hypersecretion, goblet cell metaplasia, airway hyperresponsiveness, and tissue remodeling [16,28]. Both cytokines induce chemokine production to recruit eosinophils and other effector cells while also driving M2 macrophage activation and tissue remodeling to coordinate multiple arms of type 2 immunity [27].

While TSLP, IL-4, and IL-13 are key regulators in mediating Th2 allergic inflammation, dysregulated type 2 immunity is the origin of the onset and persistence of many types of allergic and inflammatory diseases. Aberrant TSLP, IL-4, and IL-13 levels and activities in the airway lead to the onset and sustaining of atopic asthma and chronic obstructive pulmonary disease (COPD) [29,30,31]. In addition, their dysfunctional activities in the skin are linked to allergic diseases including atopic dermatitis, allergic rhinitis, and chronic rhinosinusitis with nasal polyps (CRSwNP), while dysregulation in the gut underlies food allergy and inflammatory bowel disease (IBD) [14,32,33,34,35]. In addition to its canonical role in allergic inflammation, aberrant TSLP activity has been implicated in autoimmune diseases such as rheumatoid arthritis and psoriasis [36,37,38,39], as well as in epithelial cancers [40]. However, IL-4/IL-13 are not primary drivers of most autoimmune diseases, and they can either be protective or pathogenic depending on specific autoimmune or chronic inflammatory settings [41,42].

Therapeutic antibodies targeting either TSLP or IL-4/IL-13 have validated their roles in human allergic inflammatory diseases. Tezepelumab, an anti-TSLP antibody, significantly reduced exacerbation and airway inflammation in severe asthma [43], confirming the importance of upstream epithelial alarmin signaling. Dupilumab, an IL-4Rα antagonist, has demonstrated efficacy across multiple type 2 diseases, including asthma, atopic dermatitis, CRSwNP, and eosinophilic COPD [44,45]. Tralokinumab and lebrikizumab, antibodies blocking IL-13, were approved for the treatment of severe atopic dermatitis [46,47,48]. However, neither agent showed complete efficacy. Tezepelumab failed to meet endpoints in moderate-to-severe atopic dermatitis [49], while two-thirds of patients remain partial or non-responders to dupilumab [50] and tralokinumab failed in asthma trials [51,52]. These observations underscore the heterogeneity of type 2 inflammation and the limitations of single-pathway blockade.

Given their complementary roles, TSLP initiating inflammation at epithelial barriers and IL-4/IL-13 sustaining effector responses, simultaneous blockade of TSLP and IL-4/IL-13 could present a better therapeutic strategy over just a single blockade. Dual inhibition could prevent initiation of Th2 polarization while also suppressing downstream effector pathways, thereby providing more comprehensive and durable disease control. In this regard, lunsekimig (also known as SAR443765), a bispecific nanobody-based molecule targeting both TSLP and IL-13, is being evaluated in Phase II trials for asthma [53]. Given the distinct, although overlapping, roles of both IL-4 and IL-13 in the Th2 allergic inflammation cascade, blocking both of them along with TSLP instead of just IL-13 with TSLP may offer a more comprehensive shutdown of the Th2 inflammation responses [54,55]. Indeed, PF-07275315, a trispecific antibody targeting TSLP, IL-4 and IL-13 simultaneously, is being evaluated in Phase II trials for asthma and atopic dermatitis. In addition, BsAb or combo antibodies targeting TSLP and IL-4Rα, the shared component of receptor complexes mediating the activities of both IL-4 and IL-13, are being developed [56].

In this study, we described the generation of a bispecific antibody targeting TSLP and IL-4Rα. We showed how this BsAb potently neutralized signaling from TSLP, IL-4, and IL-13; reduced inflammatory responses in human cell-based assays; and provided broader efficacy in a TSLP/OVA-induced asthma model compared with single-pathway blockade. These findings supported how dual TSLP/IL-4Rα inhibition could be a promising therapeutic approach for allergic and chronic inflammatory diseases.

## 2. Materials and Methods

### 2.1. Antibody Expression and Purification

Antibody heavy chain (HC) and light chain (LC) constructs with 1:3 molar ratio were co-transfected into Expi293F cells by the Expifectmine293 transfection kit (ThermoFisher Scientific, San Jose, CA, USA). After 5 days of incubation at 37 °C, cell supernatant was harvested, and the expressed antibody was purified over a MabSelect SuRe column followed by buffer exchange over a desalting column (Cytiva, Marlborough, MA, USA). The tezepelumab and dupilumab analogue antibodies (hitherto referred to just as tezepelumab and dupilumab, respectively) were generated in-house based on HC and LC sequences deposited in Protein Data Bank (5J13 for tezepelumab and 6WGB for dupilumab).

### 2.2. BsAb Generation by Controlled Fab Arm Exchange

The BsAb targeting TSLP and IL-4Rα was generated from anti-TSLP and anti-IL-4Rα parental antibodies in a process known as controlled Fab-arm exchange [57]. Both parental antibodies in equal molar amounts were mixed and reduced under 75 mM 2-mercaptoethylamine (2-MEA) for five hours, followed by dialysis against DPBS to allow BsAb formation. The formation of BsAb was assessed by cation exchange (CEX) high-pressure liquid chromatography over a Bio SCX column (Agilent Technologies, Santa Clara, CA, USA) under pH 6.0. The efficiency of BsAb formation was calculated by integration of the BsAb peak versus residual parental antibody peaks. The quality of BsAb was also assessed by size exclusion (SEC) high-pressure liquid chromatography over an AdvanceBio SEC column (Agilent Technologies, Santa Clara, CA, USA) and SDS-PAGE (4–12% gel, ThermoFisher Scientific, San Jose, CA, USA) under reduced and non-reduced conditions.

### 2.3. ELISA Binding Assay

Recombinant human TSLP, IL-4Rα, and IL-4Rα/IL-13Rα1 proteins (AcroBiosystems, Newark, DE, USA) at the 1 μg/mL concentration were pre-coated on 96-well Maxisorp plate (ThermoFisher Scientific, San Jose, CA, USA) overnight. After blocking, testing antibodies were added and incubated for 2 h at room temperature by shaking. After washing the plates four times, HRP-conjugated anti-human IgG secondary antibody (Jackson ImmunoResearch Laboratories, West Grove, PA, USA) was added followed by TMB (3,3′,5,5′-tetramethylbenzidine) substrate (ThermoFisher Scientific, San Jose, CA, USA) for color development. Optical densities at 450 nm wavelength were determined with the SpectraMax i3X plate reader (Molecular Devices, Sunnyvale, CA, USA). The data were plotted versus the logarithm of antibody concentrations. A four-parameter logistic (4PL) sigmoidal dose–response analysis was performed by GraphPad Prism 10.2.3 (GraphPad Software, La Jolla, CA, USA) to calculate the EC_50_ values of binding.

### 2.4. Blocking TSLP Binding to Its Cell Surface Receptor Complex

Testing antibodies along with 10 ng/mL of biotinylated human TSLP (Acro biosystems, Newark, DE, USA) were applied to Expi293F cells co-transfected with constructs expressing human IL-7Rα and TSLP and incubated for 2 h. PE-conjugated streptavidin (BioLegend, San Diego, CA, USA) was then applied and the staining of TSLP bound on the surface of transfected cells was evaluated by flow cytometry using MACSQuant Analyzer 10 (Miltenyi Biotec, Bergisch Gladbach, Germany). The normalized percentages of TSLP binding were plotted against the testing antibody concentrations. A four-parameter logistic (4PL) sigmoidal dose–response analysis was performed by GraphPad Prism 10.2.3 (GraphPad Software, La Jolla, CA, USA) to calculate the IC_50_ values of blocking TSLP binding to its cell surface receptor complex.

### 2.5. TSLP-Mediated Reporter Gene Assay

HEK293T cells were co-transfected with constructs expressing IL-7Rα, TSLPR, and pGL4.52 STAT5 luciferase reporter gene (Promega, Madison, MI, USA) by lipofectamine 3000 (ThermoFisher Scientific, San Jose, CA, USA). One day post-transfection, testing antibodies along with 3 ng/mL recombinant human TSLP (R&D systems, Minneapolis, MN, USA) were applied to the transfected cells and incubated for 24 h. The luciferase reporter gene activity was determined using the ONE-Glo™ EX Luciferase Assay kit (Promega, Madison, MI, USA). The normalized percentages of TSLP activity were plotted against the testing antibody concentrations. A four-parameter logistic (4PL) sigmoidal dose–response analysis was performed by GraphPad Prism 10.2.3 to calculate the IC_50_ values of functional neutralization.

### 2.6. TSLP-Driven Proliferation of Activated Human CD4^+^ T Cell Assay

Human CD4^+^ T cells, isolated from PBMC (Hemacare, Northridge, CA, USA) from healthy donors with the MojoSort Human CD4 T cell isolation kit (BioLegend, San Diego, CA, USA), were labelled with 5 μM Cell Proliferation Dye eFluor 450 (ThermoFisher Scientific, San Jose, CA, USA). Then the cells, plated at 2 × 10^5^ cells/well on 96-well plate pre-coated with anti-CD3 (OKT3) antibody, were treated with 50 ng/mL human TSLP and 10 μg/mL testing antibodies for six days. The labelled CD4^+^ T cells were then analyzed by flow cytometry using MACSQuant Analyzer 10 (Miltenyi Biotec, Bergisch Gladbach, Germany). Proliferation of CD4^+^ T cells was assessed by the quantitation of cell fractions with progressively diluted staining by the cell proliferation dye eFluor-450 (ThermoFisher Scientific, San Jose, CA, USA).

### 2.7. IL-4/IL-13-Mediated HEK-Blue Reporter Gene Assay

A HEK-Blue IL-4/IL-13 reporter assay was conducted with HEK-Blue IL-4/IL-13 reporter cells (InvivoGen, San Diego, CA, USA). 5 × 10^4^ cells seeded in 96-well tissue plates were treated with testing antibodies along with 1 ng/mL of IL-4 or 3 ng/mL of IL-13. After overnight incubation, the secreted embryonic alkaline phosphatase (SEAP) reporter gene activity was quantitated using a Quanti-Blue detection kit (InvivoGen, San Diego, CA, USA). The normalized percentages of IL-4/IL-13-driven reporter gene expression were plotted against the testing antibody concentrations. A four-parameter logistic (4PL) sigmoidal dose–response analysis was performed by GraphPad Prism 10.2.3 to determine the IC_50_ values of functional neutralization.

### 2.8. IL-4/IL-13-Mediated TF-1 Cell Proliferation Assay

Human myeloid progenitor TF-1 (ATCC, Manassas, VA, USA) cells, seeded at 8000 cells/well in a 96-well plate in RPMI complete growth medium excluding GM-CSF, were treated with testing antibodies and 10 ng/mL human IL-4 or IL-13. Four days after treatment, IL-4/IL-13-mediated TF-1 cell proliferation was quantitated using a CellTiter-Glo luminescent cell viability assay kit (Promega, Madison, WI, USA). The normalized percentages of IL-4/IL-13-driven TF-1 cell proliferation were plotted against the logarithm of testing antibody concentrations. A four-parameter logistic (4PL) sigmoidal dose–response analysis was performed by GraphPad Prism 10.2.3 to calculate the IC_50_ values of functional neutralization.

### 2.9. Der p Stimulated IL-5/CCL26 Release from PBMC, MRC-5, and A549 Cell Co-Culture Assay

1.5 × 10^5^ human lung fibroblast MRC-5 cells and 7.5 × 10^4^ human lung carcinoma A549 cells (ATCC, Manassas, VA, USA) were mixed in RPMI medium and cultured in a 24-well plate overnight. The next day, 1 × 10^6^ human PBMC (Hemacare, Northridge, CA, USA), 3 μg/mL *Dermatophagoides pteronyssinus* (Der p, Sino Biological, Paoli, PA, USA), and 3 μg/mL testing antibodies were added to the plate. After the cells were incubated for 6 days, the cell supernatants were concentrated by an Amicon Ultra centrifugal filter unit (Sigma, St. Louis, MO, USA). The amounts of IL-5 and CCL26 in cell supernatants were quantitated by ELISA using ELISA MAX Deluxe Set Human IL-5 (BioLegend, San Diego, CA, USA) and Human CCL26/Eotaxin-3 Quantikine ELISA Kit (R&D Systems, Minneapolis, MN, USA), respectively.

### 2.10. TSLP/IL-4-Mediated CCL17 Release from Activated Monocyte Assay

CD14^+^ monocytes were isolated from PBMC (Hemacare, Northridge, CA, USA) from healthy donors with the APC-conjugated anti-CD14 antibody (ThermoFisher Scientific, San Jose, CA, USA) by EasySep APC Positive Selection Kit II (StemCell Technologies, Vancouver, BC, Canada) according to the manufacturer’s instructions. The isolated monocytes were pretreated with 10 ng/mL bacterial lipopolysaccharide (LPS) overnight. The next day, cells were washed and then treated with combinations of 2 ng/mL IL-4 and 50 ng/mL TSLP along with serial dilutions of testing antibodies for two days. The cell supernatants were collected, and the amounts of CCL17 protein were quantitated (R&D Systems, Minneapolis, MN, USA).

### 2.11. TSLP/OVA-Induced Asthma Model Using Humanized Mice

The allergic asthma mouse study was performed using hIL4/hIL4RA/hTSLP/hTSLPR plus humanized mice developed by Biocytogen Pharmaceuticals (Beijing, China). These transgenic mice have human *IL-4*, *IL-4Rα*, *TSLP*, *TSLPR*, and *IL-7Rα* genes engineered into the genome of C57BL/6 mice to replace their murine counterparts. A total of 27 female mice with ages of 7–9 weeks and body weights of 20–22 g were randomly allocated into 5 groups based on their body weights. Mice were housed in groups under standard temperature, humidity, and a 12 h light/dark cycle, with ad libitum access to chow and water. Group A is the sham control group in which three mice were treated with 20 μL PBS by intranasal administration every two days. Six mice were allocated to each treatment group to provide statistical power for group comparisons while minimizing unnecessary animal use. The allergic asthma was induced and established by the administration of 20 μL PBS containing 40 μg OVA (ovalbumin) and 1 μg human TSLP intranasally to these mice every other day for two weeks. On Day-1 and Day 6, antibody drugs were administered via intraperitoneal injection for the drug efficacy study with the following regimen: 5 mg/kg isotype control antibody, tezepelumab, and dupilumab for group B, C, and D mice, respectively, and 10 mg/kg TAVO101 × IL4R-dupi antibody for group E mice. The study was terminated on Day 14, and serum, bronchoalveolar lavage fluid (BALF) samples, and lung tissues were collected from all mice (except one dead mouse in group D) for ELISA, flow cytometry, and lung histopathology analysis. Animals were handled by trained personnel to minimize distress. No unexpected adverse events occurred. Routine institutional monitoring was performed; no specific humane endpoints were established beyond standard guidelines. Potential confounders such as treatment order, measurement order, and cage location were not controlled, and no blinding procedures were implemented in this study.

Total IgE concentrations in the serum were determined by ELISA-based quantitation assay (BioLegend, San Diego, CA, USA). The lung samples were processed to measure the levels of mouse IL-5, CCL17, TNFα, and IFNγ cytokines by MSD U-PLEX multiplex immunoassays (MesoScale Discovery, Rockville, MD, USA). BALF from mice was collected to quantitate leukocytes, eosinophils, alveolar macrophages, and neutrophils by flow cytometry analysis. The upper lobes of right lung tissues were collected for histopathological analysis by haematoxylin and eosin (H&E) and Periodic Acid Schiff (PAS) staining. The pathological assessment includes scoring for inflammatory cell infiltration around blood vessels and bronchioles and scoring for eosinophil infiltration. The PAS-positive areas were also scored for goblet cell metaplasia and mucous production. A four grade scoring system—minimal (1), slight (2), moderate (3), and severe (4), as previously described [58]—was used to quantitate immune cell infiltration and positive cell staining. Results were represented by means and the standard error (Mean ± SEM). The data were analyzed by the One-way ANOVA with Dunnett’s multiple comparisons test (* *p* < 0.05, ** *p* < 0.01, *** *p* < 0.001, **** *p* < 0.0001) using GraphPad Prism 10.2.3 with default settings; no additional formal tests of distributional assumptions were conducted.

### 2.12. FcRn Binding Assay

Recombinant mouse FcRn (R&D Systems, Minneapolis, MN, USA) at 2 μg/mL was coated on a Maxisorp 96-well plate overnight at 4 °C. Testing antibodies in the blocking buffer containing 0.05 M MES [2-(*N*-morpholino) ethanesulfonic acid], 1% (*v*/*v*) bovine serum albumin (BSA), 0.02% (*v*/*v*) Tween-20, pH 6.0 were added to the plate and incubated for 2 h. Then HRP-conjugated anti-human IgG secondary antibody (Jackson ImmunoResearch Laboratories, West Grove, PA, USA) was added followed by the TMB substrate (ThermoFisher Scientific, San Jose, CA, USA) for color development. Optical densities at 450 nm wavelength was plotted versus the antibody concentrations. A four-parameter logistic (4PL) sigmoidal dose–response analysis was performed by GraphPad Prism 10.2.3 to determine the EC_50_ values of binding.

### 2.13. FcγRI (CD64) and FcγRIIIA (CD16a) Binding Assay

Testing antibodies were added to Maxisorp 96-well plate pre-coated with 2 μg/mL of recombinant human FcγRI (CD64) or FcγRIIIA (CD16a) (AcroBiosystems, Newark, DE, USA) and incubated for two hours. The binding of the antibodies was detected by an HRP-conjugated anti-human IgG secondary antibody (Jackson ImmunoResearch Laboratories, West Grove, PA, USA) followed by the addition of TMB substrate (ThermoFisher Scientific, San Jose, CA, USA). The absorbance data was obtained at 450 nm and plotted versus antibody concentrations. A four-parameter logistic (4PL) sigmoidal dose–response analysis was performed by GraphPad Prism 10.2.3 to determine the EC_50_ values of binding.

## 3. Results

### 3.1. Generation of a Bispecific Antibody Binding to Both TSLP and IL-4Rα

The BsAb targeting both TSLP and IL-4Rα, named TAVO101 × IL4R-dupi, comprised the anti-TSLP antibody TAVO101 Fab arm and the anti-IL-4Rα antibody dupilumab Fab arm (Figure 1A). To generate the TAVO101 × IL4R-dupi bispecific antibody, monoclonal anti-TSLP antibody TAVO101 [58] and anti-IL-4Rα antibody dupilumab (IL4R-dupi) [44] were made separately, followed by heterodimerization between the two parental antibodies through a controlled Fab-arm exchange process [57]. More specifically, F405L and K409R mutations were engineered on IgG1 Fc of TAVO101 and IL4R-dupi, respectively, to facilitate the BsAb formation. The plasmids encoding the parental antibody heavy chains and light chains were transiently co-transfected into Expi293F cells and the expressed parental antibodies were purified by affinity chromatography. For controlled Fab-arm exchange, equal molar amounts of both parental antibodies were mixed and reduced by 75 mM 2-mercaptoethylamine (2-MEA) and then dialyzed with subsequent hinge disulfide oxidation for the BsAb formation.

The formation of BsAb TAVO101 × IL4R-dupi was assessed by CEX. The profiles of peak migration for TAVO101 × IL4R-dupi, along with their respective parental antibodies TAVO101 and IL4R-dupi, were shown in Appendix A. TAVO101 × IL4R-dupi migrated as a major protein peak with the retention time in-between the migrated major peaks of the two parental antibodies, thereby indicating the formation of the BsAb. Further calculation of the protein peaks indicated that over 97% of the parental antibodies formed the desired BsAb by the Fab-arm exchange. In addition, TAVO101 × IL4R-dupi migrated as a single major protein peak without a significant level of protein aggregation analyzed by SEC (Appendix A), even after one week of storage at 37 °C. By SDS-PAGE analysis under reduced and non-reduced conditions, TAVO101 × IL4R-dupi and its associated parental antibodies showed the intact protein bands with the expected molecular weights (Appendix A).

ELISA-based binding assays evaluated the binding of TAVO101 × IL4R-dupi to recombinant human TSLP and IL-4Rα antigens. TAVO101 × IL4R-dupi (EC_50_ ~0.62 nM) had almost 5-fold less potent binding to human TSLP than TAVO101 (EC_50_ ~0.13 nM) due to the reduced valency in binding. The IL4R-dupi parental antibody did not bind to TSLP (Figure 1B). The TAVO101 × IL4R-dupi (EC_50_ ~2.63 nM) was 3-fold less potent but with higher efficacy than IL4R-dupi in binding to IL-4Rα (EC_50_ ~0.76 nM). The TAVO101 parental antibody did not bind to IL-4Rα (Figure 1C). However, both TAVO101 × IL4R-dupi (EC_50_ ~0.21 nM) and IL4R-dupi (EC_50_ 0.19 nM) bound to the IL-4Rα/IL-13Rα1 receptor complex more potently than their bindings to IL-4Rα (Figure 1D), and the difference in valency had less effect on their comparable binding.

### 3.2. Neutralization of TSLP Activity by the Bispecific Antibody TAVO101 × IL4R-dupi

The TAVO101 × IL4R-dupi blocking TSLP binding to the human TSLPR-IL-7Rα receptor complex expressed on transfected HEK293T cells was assessed by flow cytometry assay. Both TAVO101 × IL4R-dupi and TAVO101 had a concentration-dependent inhibition of human TSLP binding to the TSLPR-IL-7Rα complex (Figure 2A). TAVO101 (IC_50_ ~0.82 nM) with bivalent TSLP binding was more potent in blocking TSLP binding than TAVO101 × IL4R-dupi (IC_50_ ~3.60 nM) with monovalent TSLP binding.

A TSLP-driven STAT5 reporter gene assay assessed the functional neutralization of TSLP activity by the TAVO101 × IL4R-dupi BsAb [58]. In this assay, varying concentrations of the test articles along with 3 ng/mL recombinant human TSLP were applied to the HEK293T cells transiently transfected with TSLPR and IL-7Rα expression constructs and a STAT5-responsive luciferase reporter gene construct, and the levels of TSLP-driven luciferase reporter gene expression were quantitated 24 h later. TAVO101 × IL4R-dupi (IC_50_ ~0.25 nM) was 3-fold less potent in the neutralization of human TSLP-driven reporter gene expression than TAVO101 (IC_50_ ~0.08 nM) owing to the reduced valency in binding. IL4R-dupi did not inhibit TSLP signalling (Figure 2B).

TAVO101 × IL4R-dupi was also tested in a TSLP-driven proliferation of activated CD4^+^ T cell assay [58,59]. In this ex vivo assay, human CD4^+^ T cells isolated from two donors were labelled with Cell Proliferation Dye eFluor 450 and incubated with plate-bound anti-CD3 antibody and 50 ng/mL TSLP for six days. While TSLP stimulated the proliferation of activated CD4^+^ T cells, the addition of 10 μg/mL TAVO101 × IL4R-dupi antibody significantly neutralized TSLP activities (Figure 2C). In summary, the TSLP-binding arm of BsAb TAVO101 × IL4R-dupi demonstrated the functional neutralization of TSLP in these in vitro and ex vivo studies.

### 3.3. Neutralization of IL-4/IL-13 Activities by the Bispecific Antibody TAVO101 × IL4R-dupi

The functional neutralization of IL-4 and IL-13 activities by TAVO101 × IL4R-dupi BsAb was assessed in a HEK-Blue IL-4/IL-13 reporter assay that utilized the HEK-Blue IL-4/IL-13 cells expressing the IL-4Rα/IL-13Rα1 receptor complex and an embryonic alkaline phosphatase (SEAP) reporter gene under the control of the IFN-β minimal promoter fused to four STAT6 binding sites. In this assay, binding of IL-4 or IL-13 to its cognate receptor complex on HEK-Blue IL-4/IL-13 cells triggered a signalling cascade leading to the activation of STAT6 and the subsequent production of SEAP. Increasing amounts of testing antibodies were applied to HEK-Blue IL-4/IL-13 reporter cells stimulated with 1 ng/mL human IL-4. After overnight incubation, the SEAP reporter gene expression was quantitated via the addition of a colorimetric substrate. TAVO101 × IL4R-dupi (IC_50_ ~0.74 nM) neutralized human IL-4-stimulated reporter gene expression with a 5-fold weaker potency than the parental IL4R-dupi (IC_50_ ~0.14 nM) that was bivalent in binding IL-4Rα. The TAVO101 had no blocking activity (Figure 3A). Analogously, the test articles were screened for IL-13 activity on HEK-Blue IL-4/IL-13 reporter cells activated with 3 ng/mL human IL-13. Similarly, TAVO101 × IL4R-dupi (IC_50_ ~0.70 nM) neutralized IL-13-mediated reporter gene expression also but with 5-fold less potency than bivalent IL4R-dupi (IC_50_ ~0.13 nM) (Figure 3B).

An IL-4/IL-13-mediated TF-1 cell proliferation assay was also employed to evaluate the functional neutralization of IL-4 and IL-13 activities by TAVO101 × IL4R-dupi BsAb. The TF-1 cell is a human hemopoietic cell line whose proliferation depended on the presence of several cytokines, including IL-4 and IL-13 [60]. In this assay, test articles were applied to TF-1 cells stimulated with 10 ng/mL human IL-4, and cell proliferation was quantitated four days later. TAVO101 × IL4R-dupi (IC_50_ ~474 ng/mL or 3.27 nM) neutralized human IL-4-stimulated TF-1 cell proliferation with a 3-fold weaker potency than IL4R-dupi (IC_50_ ~170 ng/mL or 1.17 nM), while the TAVO101 had no blocking activity (Figure 3C). Likewise, the TAVO101 × IL4R-dupi (IC_50_ ~109 ng/mL or 0.75 nM) blocked 10 ng/mL human IL-13-stimulated TF-1 cell proliferation with a 2-fold weaker potency than the bivalent IL4R-dupi (IC_50_ ~50.3 ng/mL or 0.35 nM) (Figure 3D). These cell-based assays demonstrated the functional integrity of the IL-4Rα-binding arm of BsAb TAVO101 × IL4R-dupi in the neutralization of both IL-4 and IL-13 activities.

### 3.4. Neutralization of TSLP/IL-4/IL-13 Activities by BsAb TAVO101 × IL4R-dupi

A model for studying allergic or asthmatic responses employs the combination of PBMCs and MRC-5 fibroblasts to measure IL-4 and IL-13 production from immune cells in the presence of a lung fibroblast [61]. This cell co-culture assay assessed the functional activity of BsAb TAVO101 × IL4R-dupi in blocking cell-intrinsic TSLP and IL-4/IL-13 mediated cytokine releases. In this assay, 1.0 × 10^6^ human PBMCs, 1.5 × 10^5^ lung fibroblast MRC-5 cells, and 7.5 × 10^4^ lung carcinoma A549 cells were co-cultured in a 24-well plate and stimulated by 3 μg/mL allergen *Dermatophagoides pteronyssinus* (Der p). The TSLP secreted from lung fibroblast MRC-5 cells promoted the activation and maturation of dendritic cells treated with house dust mite allergen Der p, which led to type 2 immune responses and the release of Th2-type cytokines, including IL-4, IL-13, and IL-5, from the polarized CD4 T cells in PBMC. The IL-4 and IL-13 further stimulated the lung cancer cell A549 to secrete CCL26. Testing antibodies at 3 μg/mL concentration were applied to the ternary cell co-cultures, and the stimulated IL-5 and CCL26 levels in the supernatant were quantitated 6 days later. Both the anti-TSLP antibody TAVO101 and anti-IL4Rα antibody IL4R-dupi reduced IL-5 levels, with more dramatic reduction observed by TAVO101 (Figure 4A,B). The BsAb TAVO101 × IL4R-dupi led to even more dramatic reduction in IL-5 levels relative to blocking TSLP or IL-4/IL-13 alone with respective monospecific antibodies, demonstrating a synergistic effect of dual TSLP and IL-4/IL-13 blockade. On the other hand, both TAVO101 and IL4R-dupi reduced CCL26 levels, albeit with a more dramatic reduction observed with the IL4R-dupi (Figure 4A,B). Nonetheless, the BsAb TAVO101 × IL4R-dupi led to a similar reduction in CCL26 level as that by IL4R-dupi. Similar results were obtained from independent *ex vivo* assays with PBMCs from two different donors (Figure 4).

A CCL17 release assay assessed the functional activity of the BsAb TAVO101 × IL4R-dupi in blocking TSLP and IL-4 mediated CCL17 releases from activated CD14^+^ monocytes [62]. In this assay, monocytes isolated from donor PBMC were pretreated with 10 ng/mL bacterial lipopolysaccharide (LPS) overnight, followed by the treatment with combinations of 2 ng/mL IL-4 and 50 ng/mL TSLP for two days. The IL-4 stimulation led to the release of up to 31.8 pg/mL CCL17 from activated monocytes, while the addition of TSLP alone did not stimulate CCL17 release. However, the addition of TSLP with IL-4 further boosted CCL17 release to 45.6 pg/mL (Figure 5A). To study the blocking effects of antibodies, test articles were applied to monocytes activated by 2 ng/mL IL-4 and 50 ng/mL TSLP, and the amounts of CCL17 in the supernatant were quantitated. TAVO101 showed partial inhibition of CCL17 release by erasing the amount of CCL17 release up to that only induced by TSLP (Figure 5B). In contrast, IL4R-dupi had a complete blocking of CCL17 release upon stimulation by IL-4 and TSLP. The BsAb TAVO101 × IL4R-dupi also led to complete reduction in CCL17 levels but with reduced potency relative to the IL4R-dupi antibody (Figure 5B).

### 3.5. Efficacy of BsAb TAVO101 × IL4R-dupi in a TSLP/OVA-Induced Asthma Model

Since neither TAVO101 nor IL4R-dupi recognized the murine TSLP or IL-4Rα targets, respectively, the in vivo efficacy of BsAb TAVO101 × IL4R-dupi was evaluated in the hIL4/hIL4RA/hTSLP/hTSLPR plus transgenic mice in which the endogenous mouse *IL-4*, *IL-4Rα*, *TSLP*, *TSLPR*, and *IL-7Rα* genes were genetically replaced by their corresponding human sequences. The allergic asthma disease state was induced by the administration of 40 μg OVA (ovalbumin) and 1 μg human TSLP intranasally into mice in groups B, C, D, and E every other day for two weeks, while mice in group A received PBS treatment as a sham control (Figure 6A). TAVO101 × IL4R-dupi at 10 mg/kg (group E) was administered once a week via intraperitoneal injection on Days-1 and 6. Likewise, the isotype control antibody (group B), tezepelumab (group C), and dupilumab (group D) at a dose of 5 mg/kg were administered as references with the same dosing scheme (Figure 6A). On day 14, mouse serum, BALF, and lung tissues were collected for ELISA, flow cytometry, and lung histopathology analysis.

During the entire treatment period, all mice maintained normal food intake and body weights (Appendix A). These profiles showed that the test articles were well tolerated.

Total IgE levels in serum were measured on day 14. Mice in group B administered with TSLP/OVA showed increased serum IgE levels (albeit not statistically significant) when compared to mice in group A without the TSLP/OVA challenge (Figure 6B), confirming the successful establishment of the asthma model. Administration of TAVO101 × IL4R-dupi in group E resulted in a considerable reduction in serum total IgE levels when compared to the isotype control group B. Likewise, dupilumab dosed at 5 mg/kg also led to a considerable reduction in serum total IgE to a level similar to that dosed with TAVO101 × IL4R-dupi, while no significant reduction was observed with tezepelumab dosing in group C (Figure 6B).

For key cytokines, TSLP/OVA administration led to significant increases in lung IL-5 and CCL17 levels in mice in group B relative to control group A. The treatments with tezepelumab (group C), dupilumab (group D), and TAVO101 × IL4R-dupi (group E) all mitigated the increase in these two cytokines to similar levels, although statistically significant differences were only achieved for tezepelumab in reducing IL-5 (Figure 6C,D). However, the addition of tezepelumab, dupilumab, or TAVO101 × IL4R-dupi only had minimal reduction in TSLP/OVA-induced IFNγ levels and did not reduce TSLP/OVA-induced TNFα levels (Appendix A).

Flow cytometry analysis of cells isolated from BALF assessed the infiltration of inflammatory cells associated with asthma. A significant increase in the percentage of eosinophils in CD45^+^ leukocytes was observed in the BALF of mice in group B with the TSLP/OVA challenge when compared to group A without the challenge (Figure 6E). Relative to the isotype control group B, the treatments with tezepelumab (group C), dupilumab (group D), and TAVO101 × IL4R-dupi (group E) all significantly mitigated the percentage of eosinophils in CD45^+^ leukocytes in the BALF of mice, with a bit more significant mitigation effect observed for TAVO101 × IL4R-dupi relative to either tezepelumab or dupilumab (Figure 6E).

By histopathological staining and scoring of the asthmatic animal lungs, the inflammatory cell infiltration and mucus production in the bronchioles were assessed. A significant increase in the infiltration of inflammatory cells and eosinophils around blood vessels and bronchioles was observed in the lungs of group B mice relative to the control group A, indicating the establishment of asthma-related lesions by TSLP/OVA administration (Figure 6F,G). Treatments with tezepelumab (group C), dupilumab (group D), and TAVO101 × IL4R-dupi (group E) reduced (albeit not statistically significantly) the infiltrations of inflammatory cells (Figure 6F). However, only the treatment with tezepelumab or TAVO101 × IL4R-dupi reduced, although not statistically significantly, the infiltrations of eosinophils, while the IL4R-dupi treatment failed to show reduction (Figure 6G). In addition, a significant increase in mucus overproduction and goblet cell metaplasia was observed in the asthmatic lung of mice in group B upon TSLP/OVA stimulation, while such disease phenotypes were mitigated (albeit not statistically significantly) by the treatments with tezepelumab (group C), dupilumab (group D), and TAVO101 × IL4R-dupi (group E), with some stronger mitigation effects by tezepelumab and TAVO101 × IL4R-dupi relative to dupilumab (Figure 6H). These data from the murine asthma model exhibited how BsAb TAVO101 × IL4R-dupi could mitigate different markers of asthma more comprehensively than single blockade of either TSLP or IL-4Rα.

### 3.6. Fc Engineering of TAVO101 × IL4R-dupi BsAb for Half-Life Extension and Reduced Effector Functions

Mutations could be engineered on the fragment crystallizable region (Fc region) of IgG antibody to alter its Fc receptor engagement in order to improve pharmacokinetic profile with reduced potentials of Fc-mediated cytotoxicity. Specifically, M428L/N434S mutations could increase the antibody in binding to FcRn that led to the extension of antibody half-life [63]. The L234A and L235A double mutations in the lower hinge region could weaken antibody binding to Fcγ receptors thereby resulting in reduced effector functions [64]. These four mutations, L234A, L235A, M428L, and N434S, collectively designated as AALS mutations, were introduced into the IgG1 Fc of TAVO101 × IL4R-dupi BsAb.

Whether the Fc-engineered antibody has improved FcRn binding affinity was assessed in an ELISA-based binding assay. TAVO101 × IL4R-dupi with the Fc AALS mutations was applied to the ELISA plate for its binding to coated mouse FcRn under pH 6.0. As a comparison, tezepelumab with native IgG2 Fc, dupilumab with native IgG4 Fc, and a null antibody with native IgG1 Fc were also assessed. The TAVO101 × IL4R-dupi bound FcRn with higher potency and efficacy than the reference antibodies with native IgG Fc, owing to the presence of M428L and N434S mutations for half-life extension (Figure 7A).

To assess whether the Fc-engineered antibody may have reduced interactions with Fcγ receptors, the binding to human FcγRI (CD64) and FcγRIIIA (CD16a), which are the major Fcγ receptors mediating antibody-dependent cellular cytotoxicity (ADCC) and antibody-dependent cellular phagocytosis (ADCP) activities, by TAVO101 × IL4R-dupi with AALS mutations was assessed in ELISA-based binding assays. TAVO101 × IL4R-dupi antibody showed minimal binding to FcγRI (CD64) and FcγRIIIA (CD16a), respectively, when compared to a null antibody with native IgG1 Fc (Figure 7B,C).

## 4. Discussion

Aberrant activities of cytokines participating in the allergic Th2 inflammation cascade, including the upstream epithelial alarmin sensing environmental insults represented by TSLP and downstream mediator and effector types of Th2 cytokines, including IL-4, IL-5, and IL-13, are the leading causes of complex allergic inflammatory diseases, including atopic asthma, COPD, atopic dermatitis, CRSwNP, food allergy, and IBD. In this study, we generated and characterized a BsAb TAVO101 × IL4R-dupi targeting both TSLP and IL-4Rα, the shared receptor component that mediates both IL-4 and IL-13 activities. By blocking both the initiator and major effectors of the Th2 inflammation cascade, we hypothesized that the BsAb could offer better disease control than single-target blockade.

The TSLP-targeting arm of our BsAb was derived from TAVO101, a humanized anti-TSLP antibody that was in clinical trial investigations for TSLP-mediated allergic inflammatory diseases [58,65]. TAVO101 × IL4R-dupi was shown to potently inhibit TSLP binding to its cell surface receptor complex and neutralize TSLP-mediated reporter gene activation as well as CD4^+^ T cell proliferation. On the other hand, the IL-4Rα targeting arm of TAVO101 × IL4R-dupi was derived from dupilumab, an anti-IL-4Rα antagonist antibody approved for the treatments of several allergic inflammatory disorders [44,45]. The TAVO101 × IL4R-dupi potently neutralized both IL-4 and IL-13-driven reporter signaling and TF-1 cell proliferation. These cell-based functional assays validated the functional integrities of both TSLP and IL-4Rα targeting arms in BsAb format. However, the BsAb TAVO101 × IL4R-dupi did not neutralize TSLP and IL-4/IL-13 activities as potently as the corresponding monospecific antibodies TAVO101 and IL4R-dupi, with roughly 3-5-fold drops in potencies, likely due to the monovalent target binding in the BsAb format.

Besides neutralization of either TSLP or IL-4/IL-13 activity, the BsAb TAVO101 × IL4R-dupi also inhibited the activities of TSLP plus IL-4 in a CCL17 release assay using activated CD14^+^ monocytes. In this assay, LPS treatment elicited CD14^+^ monocytes to express TSLP receptor complex components TSLPR and IL-7Rα to enable TSLP synergy with IL-4 to enhance the production of chemokine CCL17 [62]. By shutting down the downstream effector cytokine IL-4, the IL4R-dupi antibody had a complete inhibition of CCL17 production. On the other hand, by blocking the initiator, TAVO101 antibody neutralized a portion of CCL17 production, which was enhanced by TSLP stimulation. By the neutralization of the activities of both the initiator and effector, the BsAb TAVO101 × IL4R-dupi also led to a complete reduction in CCL17 levels. However, due to monovalent IL-4 binding, the BsAb was not as potent as the bivalent IL4R-dupi antibody in the inhibition of CCL17 release.

In a more complex *ex vivo* allergen-stimulated cytokine release assay using the co-culture of PBMC, MRC-5, and A549 cells, the BsAb targeting both TSLP and IL-4Rα showed synergy of dual inhibition. In this assay, neither TSLP nor IL-4/IL-13 was added extrinsically; instead, they were produced autonomously by the cells. TSLP, secreted from lung fibroblast MRC-5 cells, synergized with allergen Der p to activate dendritic cells and then polarized CD4^+^ T cells to release Th2-type cytokines, including IL-4, IL-13, and IL-5. The IL-4 and IL-13 stimulated the lung cancer cell A549 to secrete CCL26 [61]. Consistent with this mainstream signalling cascade, TAVO101 blocked the activation of the TSLP pathway, thereby significantly inhibiting IL-5 production. The IL4R-dupi blocked the IL-4/IL-13 signalling pathway, thereby significantly inhibiting CCL26 secretion. However, neither TAVO101 nor IL4R-dupi could completely inhibit the release of IL-5 and CCL26, respectively, suggesting the existence of a complementary cascade controlling the release of these cytokines. Related, the observation that blocking IL-4/IL-13 by IL4R-dupi antibody also led to some inhibition of IL-5 production hinted at a feedback loop regulation of IL-5 production driven by IL-4/IL-13. These results were consistent with previous reports of complementary and interconnected biology among these Th2 cytokines. While TSLP-activated dendritic cells promote IL-4 secretion by naïve T cells [3,5], IL-4/IL-13 signalling can enhance epithelial alarmin expression, including TSLP, thereby fuelling a self-reinforcing circuit [8,14]. Separate studies showed that IL-13 stimulation of airway epithelial cells upregulates TSLP, amplifying the barrier cytokine responses [66,67]. Given the bidirectional feedback loop and complementary regulation among these Th2 cytokines, it was remarkable to observe that the BsAb TAVO101 × IL4R-dupi nearly completely inhibited IL-5 secretion, which was more efficacious than blocking TSLP or IL-4/IL-13 alone by respective monospecific antibodies. TAVO101 × IL4R-dupi also led to a similar reduction in CCL26 level as that by IL4R-dupi. The BsAb synergistic effects demonstrated that single pathway blockade led to potential escape routes, while the dual inhibition of both initiation and effector arms of type 2 inflammation disrupted the cycle leading to cooperative suppression.

The synergy of dual inhibition of TSLP and IL-4Rα was also evident phenotypically in the humanized mouse asthma model. *In vivo*, the BsAb TAVO101 × IL4R-dupi reduced all hallmark features of allergic disease, including IgE production, IL-5 and CCL17 levels, eosinophilic infiltration, and mucus hypersecretion. In contrast, blocking TSLP alone by tezepelumab failed to reduce the IgE production, while blocking IL-4Rα alone by dupilumab did not affect the infiltration of eosinophils into the lesion lung and also led to less reduction in mucus hypersecretion. The more comprehensive and consistent suppression of type 2 inflammation by dual inhibition than monotherapies reflected the distinct while cooperative roles between TSLP and IL-4/IL-13. TSLP, released by epithelial cells under allergen or environmental stress, primed dendritic cells and initiated Th2 differentiation to amplify allergic inflammation [1,2]. IL-4 and IL-13, acting via IL-4Rα, sustain effector responses such as IgE class switching, eosinophil recruitment, and airway remodeling [17,27]. However, while IL-4 and IL-13 are the major effectors, TSLP also amplifies other Th2 cytokines, such as IL-5, to mediate its function. On the other hand, IL-4/IL-13 signaling can enhance the expression of other epithelial alarmins besides TSLP, such as IL-33, to fuel a self-reinforcing loop [8,14]. This complex interconnected biology among these Th2 cytokines could explain why a single blockade could not reduce certain allergic biomarkers, while the dual blockade of both TSLP as the initiator and IL-4/IL-13 as central mediators could provide a more comprehensive control of all aspects and stages of allergic disease progression. Nonetheless, whether the dual blockade could have better control of markers related to tissue remodeling and oxidative stress, and pulmonary function remains to be explored. Considering the many differences between humans and mice that include tissue immune cell distributions and pharmacodynamics, confirmation of the translation of the efficacies of dual inhibition observed in this asthma model remains to be seen. More animal efficacy studies would also be needed to explore whether the BsAb targeting TSLP and IL-4Rα at higher dosing levels could have synergistic effects besides comprehensively controlling disease progression. In addition, whether the BsAb could have efficacies in other allergic inflammatory disease models and possible safety concerns due to excessive immunosuppression remain to be explored.

These mechanistic insights aligned with outcomes observed in patients treated with single-pathway biologics. Dupilumab, an IL-4Rα antagonist, provided substantial benefit in asthma and atopic dermatitis, yet 30–40% of patients remained partial or non-responders [44,45]. Tezepelumab, targeting TSLP, significantly reduced exacerbations in severe asthma [43] but failed to meet endpoints in moderate-to-severe atopic dermatitis [29]. Our data suggested that dual TSLP/IL-4Rα inhibition could overcome single blockade shortcomings by collapsing the cytokine feedback loop between epithelial alarmins and Th2 effector cytokines, thereby achieving broader and more consistent clinical responses. Clinically, this could translate into reduced exacerbation frequency, improved lung function, and higher responder rates in asthma, as well as deeper and more durable improvement in atopic dermatitis and chronic rhinosinusitis with nasal polyps. Indeed, multiple bispecific or trispecific antibodies targeting TSLP in combination with targeting IL-13 alone, IL-4 and IL-13 together, or the IL-4Rα receptor are being developed to test this idea [53,54,55,56]. Despite sharing many common regulatory pathways and receptors, IL-4 and IL-13 have differential expression due to distinct cellular sources and tissue localization and perform very distinct functions during a type-2 immune response [68,69]. Recent studies indicated broader blockade of Th2 inflammation in disease states by blocking IL-4 and IL-13 together than by blocking IL-13 alone [51,54,55], suggesting that blocking both cytokines should be combined with blocking TSLP. However, whether blocking these cytokines would have safety and tolerability concerns in clinical practice due to excessive immunosuppression remains to be explored, although dual blockade of TSLP and IL-13 by lunsekimig or dual blockade of IL-33 and IL-4Rα by a combination of itepekimab and dupilumab were well tolerated with acceptable safety profiles in clinical trials [53,70].

Beyond efficacy, Fc engineering conferred favorable pharmacologic properties. The incorporation of the AALS mutations in our TAVO101 × IL4R-dupi BsAb reduced Fcγ receptor binding, minimizing effector function–related risks such as antibody-dependent cytotoxicity, while enhancing FcRn binding to extend half-life. Such improvements are particularly valuable for chronic conditions requiring long-term therapy, where extended dosing intervals and improved safety could enhance adherence and clinical outcomes. However, due to target-mediated drug disposition (TMDD), dupilumab sustained activity could suffer from a limited half-life [71]. It remains to be explored in a PK study in a relevant animal model as to whether our TAVO101 × IL4R-dupi BsAb engineered with FcRn binding-enhancing mutations could mitigate the quicker drug disposition due to the targeting of cell surface receptors composed of IL-4Rα. Alternatively, targeting the soluble IL-4 and IL-13 cytokines directly rather than their cell surface receptors could be advantageous from a pharmacokinetics perspective.

## 5. Conclusions

In summary, this TAVO101 × IL4R-dupi BsAb represented a next-generation therapeutic approach that combined two validated mechanisms: inhibition of the epithelial cytokine TSLP and blockade of IL-4/IL-13 signaling via IL-4Rα. By targeting both the initiation and maintenance phases of type 2 inflammation, dual blockade could achieve more comprehensive control of allergic and chronic inflammatory diseases than existing single-target biologics. These findings provided a strong rationale for advancing such BsAb into clinical development for severe asthma, atopic dermatitis, and other related disorders where residual disease activity remained a major unmet need.

## Figures and Tables

**Figure 1 cells-14-01747-f001:**
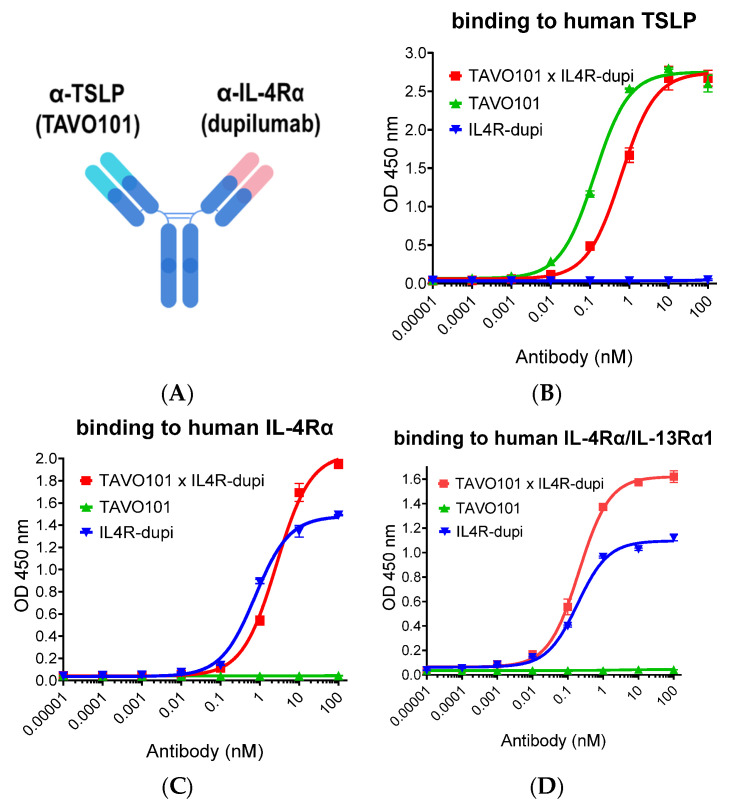
Antigen-binding properties by antibodies targeting TSLP and IL-4Rα. (**A**) Schematic drawing of BsAb TAVO101 × IL4R-dupi. (**B**–**D**). TAVO101 × IL4R-dupi, TAVO101 and IL4R-dupi antibodies were assessed for their concentration-dependent binding to immobilized human TSLP (**B**), human IL-4Rα (**C**), and human IL-4Rα/IL-13Rα1 receptor complex (**D**) in ELISA assays. Dose response curves showed ODs at 450 nm over the concentrations of testing antibodies (data expressed as mean ± SEM, n = 2).

**Figure 2 cells-14-01747-f002:**
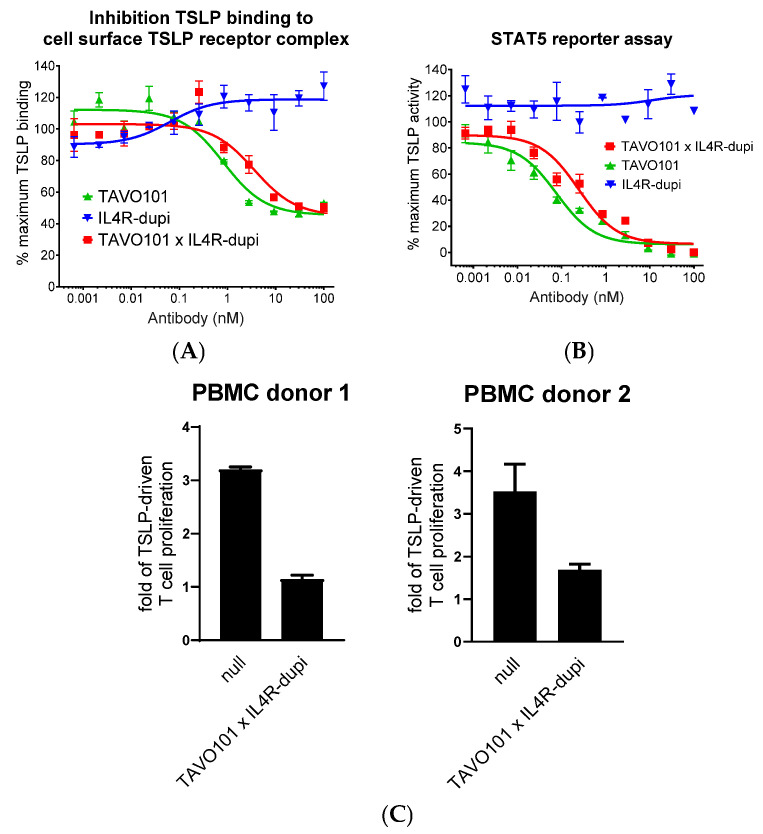
Neutralization of TSLP activity by antibodies targeting TSLP and IL-4Rα. (**A**). Neutralization of TSLP binding to its cell surface receptor complex by TAVO101 × IL4R-dupi, TAVO101, and IL4R-dupi antibodies in flow cytometry assays. Dose response curves showed the percentages of TSLP binding, normalized to the maximum binding by 10 ng/mL human TSLP, over the concentrations of test antibodies (data expressed as mean ± SEM, n = 3). (**B**). Neutralization of human TSLP-driven STAT5 reporter gene activation by TAVO101 × IL4R-dupi, TAVO101, and IL4R-dupi antibodies in the STAT5 reporter assay. Dose response curves showed the percentages of TSLP activity, normalized to the maximum activity driven by 3 ng/mL human TSLP, over the concentrations of testing antibodies (data expressed as mean ± SEM, n = 4). (**C**). Neutralization of TSLP-driven proliferation of activated human CD4^+^ T cells by TAVO101 × IL4R-dupi and null antibodies in T cell proliferation assay. Bar graphs showed folds of T cell proliferation over that without TSLP treatment. (data expressed as mean ± SEM, n = 2).

**Figure 3 cells-14-01747-f003:**
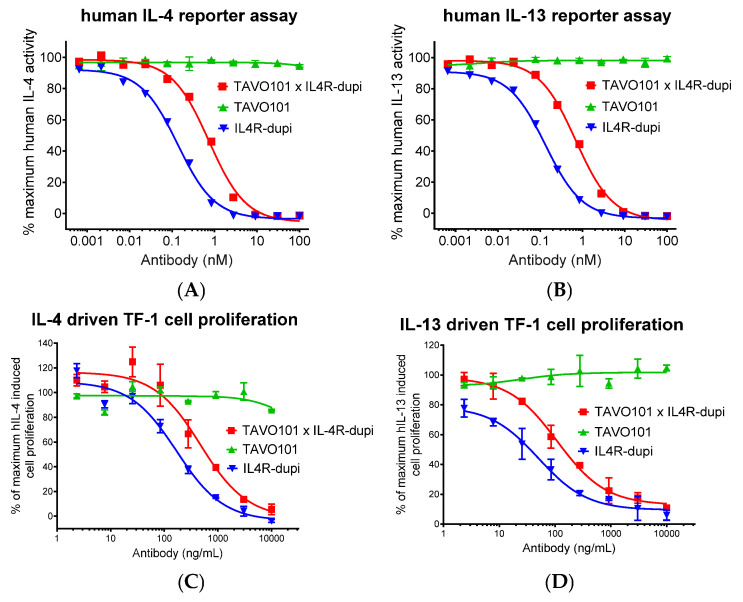
Neutralization of IL-4/IL-13 activities by antibodies targeting TSLP and IL-4Rα. (**A**,**B**) Neutralization of human IL-4 (**A**) and IL-13 (**B**) driven reporter gene activation by TAVO101 × IL4R-dupi, TAVO101, and IL4R-dupi antibodies in HEK-Blue IL-4/IL-13 reporter assays. Dose response curves showed the percentages of IL-4/IL-13 activity, normalized to the maximum activity driven by 1 ng/mL human IL-4 or 3 ng/mL IL-13, over the concentrations of testing antibodies (data expressed as mean ± SEM, n = 2). (**C**,**D**). Neutralization of human IL-4 (**C**) and IL-13 (**D**) driven proliferation of TF-1 cells by TAVO101 × IL4R-dupi, TAVO101, and IL4R-dupi antibodies in TF-1 cell proliferation assays. Dose response curves showed the percentages of IL-4/IL-13 activity, normalized to the maximum activity driven by 10 ng/mL human IL-4 or 10 ng/mL IL-13, over the concentrations of testing antibodies (data expressed as mean ± SEM, n = 2).

**Figure 4 cells-14-01747-f004:**
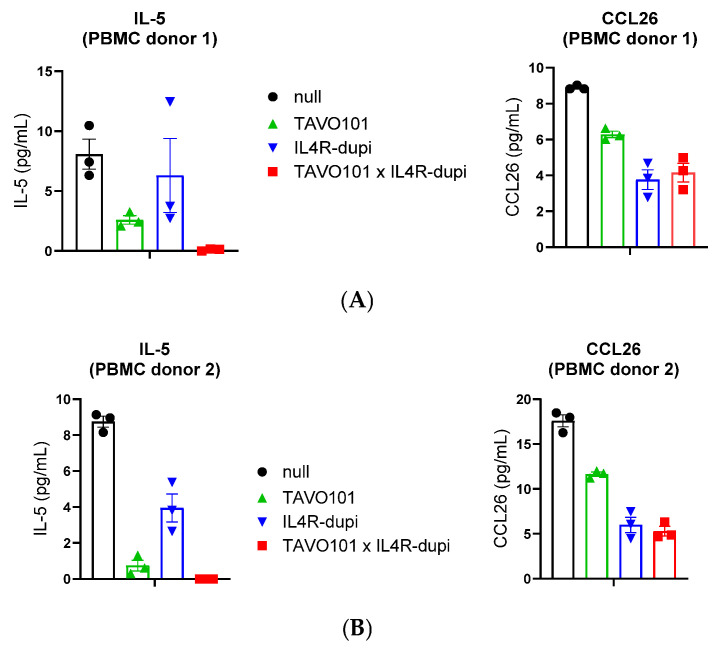
Inhibition of Der p stimulated IL-5/CCL26 release from PBMC, MRC-5, and A549 cell co-cultures by null control, TAVO101 × IL4R-dupi, TAVO101, and IL4R-dupi antibodies. Bar graphs showed the amounts of IL-5 and CCL26 released in cell supernatants. (data expressed as mean ± SEM, n = 3). The ex vivo assays were performed with PBMC from two different donors (**A**,**B**).

**Figure 5 cells-14-01747-f005:**
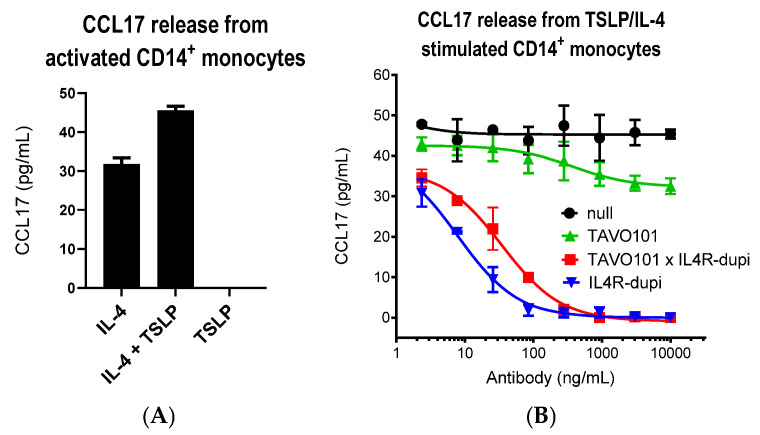
Neutralization of TSLP/IL-4-driven CCL17 release from activated CD14^+^ monocytes. (**A**). CCL17 releases from activated CD14^+^ monocytes stimulated with IL-4 alone, TSLP and IL-4, and TSLP alone. The amounts of CCL17 released in cell supernatants were plotted against the testing cytokines in the bar graph as shown (data expressed as mean ± SEM, n = 6). (**B**). Neutralization of TSLP/IL-4-driven CCL17 release from activated CD14^+^ monocytes by null control, TAVO101 × IL4R-dupi, TAVO101, and IL4R-dupi antibodies. Dose response curves showed the amounts of CCL17 released over the concentrations of testing antibodies (data expressed as mean ± SEM, n = 2).

**Figure 6 cells-14-01747-f006:**
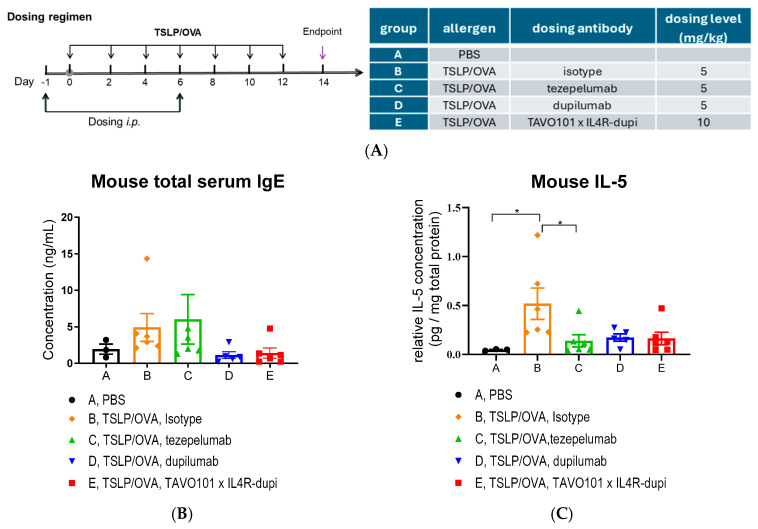
Efficacy of antibodies targeting TSLP and IL-4Rα in the TSLP/OVA-induced asthma model using hIL4/hIL4RA/hTSLP/hTSLPR plus transgenic mice. (**A**). Dosing regimen and animal grouping in the asthmatic model. Three mice were enrolled in group A, while six mice were enrolled in the asthma induction groups B to E. (**B**–**D**). Bar graphs showed the concentrations of total serum IgE (**B**), lung tissue IL-5 (**C**), and CCL17 (**D**) in mice from study groups. (**E**). Bar graphs showed the percentages of eosinophils in CD45^+^ leukocytes in BALF of asthmatic mice from study groups. (**F**–**H**). Scores of inflammatory cell infiltration (**F**), eosinophil infiltration (**G**), and goblet cell metaplasia and mucous production (**H**) of asthmatic lungs in mice from study groups. Note: Data was represented by mean ± SEM and analyzed by one-way ANOVA with Dunnett’s multiple comparisons test. Each experimental group was compared to group B (* *p* < 0.05, ** *p* < 0.01, *** *p* < 0.001, **** *p* < 0.0001).

**Figure 7 cells-14-01747-f007:**
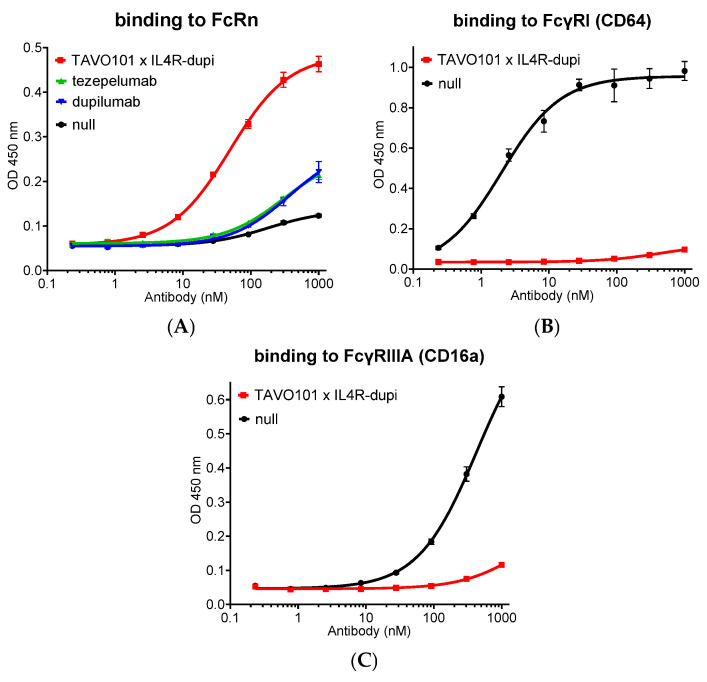
Fc engineering of TAVO101 × IL4R-dupi for extended half-life and reduced Fcγ receptor binding. (**A**). Binding to immobilized mouse FcRn at pH 6.0 by TAVO101 × IL4R-dupi, tezepelumab, dupilumab, and null antibodies in ELISA binding assays. Dose response curves showed OD at 450 nm over the concentrations of test antibodies (data expressed as mean ± SEM, n = 2). (**B**,**C**). Binding to FcγRI (CD64) (**B**) and FcγRIIIA (CD16a) (**C**) by TAVO101 × IL4R-dupi and null antibodies in ELISA binding assays. Dose response curves showed OD at 450 nm over the concentrations of test antibodies (data expressed as mean ± SEM, n = 2).

## Data Availability

All generated data are presented in the article/Appendix A. Further inquiries can be directed to the corresponding author.

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
