# Peer review of "A Bispecific Antibody Blocking Both TSLP and IL-4Rα for the Treatment of Allergic Inflammatory Diseases"

_cells, 2025, doi:10.3390/cells14221747_

Round 1

Reviewer 1 Report

Comments and Suggestions for Authors This is a well-written and methodologically sound manuscript presenting the development and preclinical validation of a bispecific antibody (BsAb) simultaneously targeting TSLP and IL-4Rα. The work is timely and relevant to the ongoing efforts to improve therapeutic strategies for type 2 inflammatory diseases such as asthma and atopic dermatitis. The study is logically designed, employing a combination of in vitro, ex vivo, and in vivo approaches to demonstrate that the BsAb (TAVO101 × IL4R-dupi) effectively blocks TSLP, IL-4, and IL-13 signalling and provides broader suppression of allergic inflammation than single-target antibodies. The methods are detailed, the data presentation is clear, and the results support the conclusions. The authors demonstrated a clear generation and validation of bispecific antibodies through biochemical and functional assays. In vivo efficacy data in a fully humanized cytokine receptor mouse model further validated the overall conclusions. The authors also offered a Comprehensive discussion linking molecular mechanisms to clinical relevance.

Major Comments

  1. Novelty and Contextualization

The Discussion summarises the dual blockade rationale, but it lacks a deeper mechanistic exploration of why TSLP and IL-4Rα co-targeting yields synergistic suppression. Do the authors have any potential explanation/hypothesis?

The manuscript provides convincing evidence that simultaneous blockade of TSLP and IL-4/IL-13 pathways offers enhanced efficacy. The introduction could be slightly expanded to compare this bispecific format with other dual or trispecific approaches currently in clinical development, emphasising what is unique in this specific case beyond the dual-target rationale.

  1. In Vivo Model Limitations

The humanized TSLP/IL-4/IL-4Rα mouse model is appropriate; however, the authors should briefly discuss potential translational limitations (e.g., differences in pharmacodynamics or immune-cell distribution compared to human tissues) to provide a balanced interpretation relevant to translation to human applications.

  1. PK data
The in vivo efficacy of the bispecific antibody is tested in humanised mice. Still, there is no information about pharmacokinetics (PK), i.e., how the antibody behaves over time, and no safety/tolerability profiling beyond general observations. Do the authors have any PK data?  

Minor Comments

Conflict of Interest Statement

The manuscript appropriately states that all authors are employees of Tavotek Biotherapeutics. It would be advisable to add a sentence affirming that the study was internally funded and that the authors had complete control over data generation and interpretation. Is this assumption correct and applicable to the specific case? Do any of the authors hold patents or equity related to the molecule? If so, a sentence should be added in the conflict of interest statement.

Cell Line Authentication

Since several HEK293T-based and reporter systems are used, it is essential to confirm the absence of mycoplasma contamination and the authentication of the cell lines.

Author Response

Comment 1: Novelty and Contextualization

 The Discussion summarises the dual blockade rationale, but it lacks a deeper mechanistic

 exploration of why TSLP and IL-4Rα co-targeting yields synergistic suppression. Do the

 authors have any potential explanation/hypothesis?

 The manuscript provides convincing evidence that simultaneous blockade of TSLP and IL

4/IL-13 pathways offers enhanced efficacy. The introduction could be slightly expanded to

 compare this bispecific format with other dual or trispecific approaches currently in clinical

 development, emphasising what is unique in this specific case beyond the dual-target

 rationale.

Response 1: We provided deeper mechanistic explanations of synergistic suppression by co-targeting TSLP and IL-4Ra in the Discussion section. On lines 750-763, we hypothesized that the synergy might be due to the overcoming self-reinforcing feedback loop regulation between these cytokines. On lines 776-783, we hypothesized that the synergy might also be due to the comprehensive control of complex signaling pathways other than the TSLP- IL-4/IL-13 pathway.

We expanded the comparison of TSLP x IL-4Ra bispecific antibody to other bi- or trispecific antibodies currently in clinical trials in a couple of paragraphs in the Discussion section instead of the Introduction section for a better flow of the manuscript. We introduced the different formats of antibodies on lines 114-124 in the Introduction section. We discussed the possible benefit of blocking both IL-4 and IL-13 by anti-IL-4Ra antibody or anti-IL-4 and IL-13 dual antibodies than by blocking IL-13 alone on lines 806-811 of the Discussion section. We also discussed the possible implications on PK by the anti-IL-4Ra antibody or anti-IL-4 and IL-13 dual antibodies on lines 822-828 of the Discussion section.

Comment 2: In Vivo Model Limitations

 The humanized TSLP/IL-4/IL-4Rα mouse model is appropriate; however, the authors

 should briefly discuss potential translational limitations (e.g., differences in

 pharmacodynamics or immune-cell distribution compared to human tissues) to provide a

balanced interpretation relevant to translation to human applications.

Response 2: Thanks to the reviewer to point this out. We added a sentence on lines 785-787 to discuss this: Considering the many differences between humans and mice that include tissue immune cell distributions and pharmacodynamics, confirmation of the translation of the efficacies of dual inhibition observed in this asthma model remains to be seen.

Comment 3: PK data

 The in vivo efficacy of the bispecific antibody is tested in humanised mice. Still, there is no

 information about pharmacokinetics (PK), i.e., how the antibody behaves over time, and

 no safety/tolerability profiling beyond general observations. Do the authors have any PK

 data? 

Response 3: We agree with the reviewer regarding the importance of PK data for TSLP x IL-4Ra bispecific antibody and we discussed this in the last paragraph of the Discussion section. Unfortunately, we do not have PK data for this proof-of-concept molecule. We are in the process of developing a proprietary TSLP x IL-4Ra bispecific molecule that will address the PK issue more carefully in future studies.

Comment: Conflict of Interest Statement

 The manuscript appropriately states that all authors are employees of Tavotek

 Biotherapeutics. It would be advisable to add a sentence affirming that the study was

 internally funded and that the authors had complete control over data generation and

 interpretation. Is this assumption correct and applicable to the specific case? Do any of the

 authors hold patents or equity related to the molecule? If so, a sentence should be added

 in the conflict of interest statement. 

Response: We added a sentence “The funders had no role in the design, execution, interpretation, or writing of the study.” in the Conflict of Interest Statement to reflect the facts that the study was internally funded and that the authors had complete control over data generation and interpretation. No authors hold patents or equity related to this proof-of-concept molecule.

Comment: Cell Line Authentication

Since several HEK293T-based and reporter systems are used, it is essential to confirm the

 absence of mycoplasma contamination and the authentication of the cell lines. 

Response: The HEK-Blue IL-4/IL-13 reporter cell line was a commercial product from Invivogen. The TSLP reporter assay was conducted on transiently-transfected HEK293T cells that were obtained from ATCC. The vendors guaranteed the authentication of these commercial cell lines and we routinely verified the cells maintained as mycoplasma-free.

Reviewer 2 Report

Comments and Suggestions for Authors

The authors conducted a well-executed study in which they described the development of a bispecific antibody targeting TSLP and IL-4Rα. They demonstrated its effects on inflammation control using human cell-based assays and a TSLP/OVA-induced asthma animal model.

Comments:

  1. The authors should provide appropriate references for the animal model employed. Additionally, the rationale for terminating the study on Day 14 should be clarified.

  2. Although histological examinations were performed, the methods used for quantifying tissue alterations need to be described in greater detail. Furthermore, the absence of assessments for other markers related to tissue remodeling and oxidative stress response should be addressed and justified.

  3. The study lacks a functional pulmonary evaluation, which would help demonstrate the impact of inflammatory control on respiratory function. The authors should clarify why such assessments were not included.

Comments on the Quality of English Language
  1. Minor grammatical and spelling corrections are recommended to improve overall clarity and readability.

Author Response

Comment 1: The authors should provide appropriate references for the animal model employed. Additionally, the rationale for terminating the study on Day 14 should be clarified.

Response 1: To our knowledge, we are the first to report the use of such OVA/TSLP-induced asthma model using the human TSLP, TSLPR, IL-7R, IL-4 and IL-4Ra transgenic mice recently developed by our vendor Biocytogen.  Hence, there is no appropriate reference to quote although there were many references for OVA-induced asthma model using wild-type mice. The study termination at Day 14 was advised by Biocytogen based on their prior experiences in model development. Besides, in our previous study (reported in reference 58) using OVA-induced asthma model on human TSLP, TSLPR and IL-7R transgenic mice, we successfully observed the treatment effects of our anti-TSLP antibody TAVO101 by terminating the study at Day 14.

Comment 2: Although histological examinations were performed, the methods used for quantifying tissue alterations need to be described in greater detail. Furthermore, the absence of

 assessments for other markers related to tissue remodeling and oxidative stress

 response should be addressed and justified.

Response 2: Thanks for the suggestion by the reviewer. We modified lines 307-312 in the Method section and referenced the quantitation methods described in greater detail in our previous study reported in Reference 58.

We agree with the reviewer that the assessments of markers related to tissue remodeling and oxidative stress response are important. However, due to the resource/funding constraints on this complicated animal model with so many disease markers being measured, we did not measure the markers for tissue remodeling and oxidative stress. We noted this limitation of the animal study on lines 783-785 in the Discussion section.

Comment 3: The study lacks a functional pulmonary evaluation, which would help demonstrate the impact of inflammatory control on respiratory function. The authors should clarify why such assessments were not included.

Response 3: We appreciate the reviewer for the suggestion of adding the evaluation of pulmonary function. However, due to the resource/funding constraints, we did not evaluate the pulmonary function. We noted this limitation of the animal study on lines 783-785 in the Discussion section.

Comment: Quality of English Language

Minor grammatical and spelling corrections are recommended to improve overall clarity

 and readability.

Response: Thanks for the reviewer’s suggestion. We hence made many grammatical corrections to improve the quality of the manuscript.

Reviewer 3 Report

Comments and Suggestions for Authors

The manuscript entitled “Bispecific antibody blocking both TSLP and IL-4Rα for the treatment of allergic inflammatory diseases” presents the design, generation, and functional characterization of a bispecific antibody (BsAb) that simultaneously targets TSLP and IL-4Rα. The authors show that this BsAb effectively neutralizes signaling and inflammatory responses mediated by TSLP, IL-4, and IL-13, both in vitro and in a humanized mouse model of asthma. The topic is timely and highly relevant to the development of next-generation biologics for allergic and chronic inflammatory diseases. Overall, the study is well designed and clearly written. However, several points should be addressed to further strengthen the manuscript before publication:

  • While the manuscript briefly discusses the complementary roles of TSLP and IL-4/IL-13, the biological rationale for targeting IL-4Rα rather than IL-13 (as adopted in other bispecific antibodies such as lunsekimig) could be further elaborated. Providing a more detailed comparison with existing bispecific or trispecific antibodies (like PF-07275315) would help contextualize the novelty and highlight the potential advantages of the proposed approach.
  • The dual blockade of upstream (TSLP) and downstream (IL-4/IL-13) pathways might increase the risk of excessive immunosuppression. The discussion should briefly address potential safety or tolerability concerns, referencing available clinical data from similar multi-target biologics.
  • The dual blockade of upstream (TSLP) and downstream (IL-4/IL-13) pathways could potentially increase the risk of excessive immunosuppression. The discussion should briefly address possible safety and tolerability concerns (both for the mouse model and for the human use).

  • The authors mention the introduction of AALS mutations and other Fc modifications but do not discuss their potential impact on effector functions (ADCC/CDC) or FcRn binding. Including a brief rationale for these mutations, along with an explanation of their expected pharmacokinetic and immunological consequences, would enhance the clarity and completeness of the manuscript.

  • Ensure consistent definitions of abbreviations across figures and text (like TARC/CCL17, PBMC, BALF).

If the authors refine these details, providing the requested explanations, and frame their discussion within an animal-to-human translational perspective, the manuscript could be published and would enrich the collective knowledge on the field.

Author Response

Comment 1: While the manuscript briefly discusses the complementary roles of TSLP and IL-4/IL-13, the biological rationale for targeting IL-4Rα rather than IL-13 (as adopted in other

 bispecific antibodies such as lunsekimig) could be further elaborated. Providing a more

 detailed comparison with existing bispecific or trispecific antibodies (like PF-07275315)

 would help contextualize the novelty and highlight the potential advantages of the

 proposed approach.

Response 1: We discussed the possible benefit of blocking both IL-4 and IL-13 by anti-IL-4Ra antibody than by blocking IL-13 alone on lines 806-811 of the Discussion section. For the comparison of TSLP x IL-4Ra bispecific antibody to other bi- or trispecific antibodies currently in clinical trials, we introduced the different formats of antibodies on lines 114-124 in the Introduction section. We argued that blocking both IL-4 and IL-13 could have advantage in disease control than by blocking IL-13 alone on lines 806-811 of the Discussion section. We also discussed the possible implications on PK from the use of the anti-IL-4Ra antibody or anti-IL-4 and IL-13 dual antibodies on lines 822-828 of the Discussion section.

Comment 2: The dual blockade of upstream (TSLP) and downstream (IL-4/IL-13) pathways might increase the risk of excessive immunosuppression. The discussion should briefly address

 potential safety or tolerability concerns, referencing available clinical data from similar

 multi-target biologics.

Response 2: We thank the reviewer for this important suggestion. We added a sentence on lines 811-815 of the Discussion section to point out this safety and tolerability concern by referencing the outcomes from two clinical trials from similar multi-target biologics.

Comment 3: The dual blockade of upstream (TSLP) and downstream (IL-4/IL-13) pathways could potentially increase the risk of excessive immunosuppression. The discussion should

 briefly address possible safety and tolerability concerns (both for the mouse model and for

 the human use).

Response 3: We thank the reviewer for this important suggestion. We pointed out this safety and tolerability concern for the mouse model (line 790-792) and for the human use (lines 811-815).

Comment 4: The authors mention the introduction of AALS mutations and other Fc modifications but do not discuss their potential impact on effector functions (ADCC/CDC) or FcRn binding. Including a brief rationale for these mutations, along with an explanation of their expected pharmacokinetic and immunological consequences, would enhance the clarity and

 completeness of the manuscript.

Response 4: On lines 660-667, we introduced these AALS mutations with their effects on Fc receptors and the associated consequences on the decreased modulations on effector functions and long-acting PK properties.

Comment 5: Ensure consistent definitions of abbreviations across figures and text (like TARC/CCL17, PBMC, BALF).

Response 5: We thank the reviewer for this suggestion. We made changes to ensure consistent definitions of the abbreviations.

Round 2

Reviewer 3 Report

Comments and Suggestions for Authors

The authors have addressed all the questions and issues I had raised. The manuscript can now be accepted and is ready for publication.